# A Scoping Review to Find Out Worldwide COVID-19 Vaccine Hesitancy and Its Underlying Determinants

**DOI:** 10.3390/vaccines9111243

**Published:** 2021-10-25

**Authors:** Md. Rafiul Biswas, Mahmood Saleh Alzubaidi, Uzair Shah, Alaa A. Abd-Alrazaq, Zubair Shah

**Affiliations:** College of Science and Engineering, Hamad Bin Khalifa University, Qatar Foundation, Doha 34110, Qatar; maal28902@hbku.edu.qa (M.S.A.); uzsh31989@hbku.edu.qa (U.S.); aabdalrazaq@hbku.edu.qa (A.A.A.-A.)

**Keywords:** PRISMA, scoping review, determinants, hesitancy, acceptance, vaccine efficacy, vaccine safety, coverage, SARS-COV2, COVID-19

## Abstract

Background: The current crisis created by the coronavirus pandemic is impacting all facets of life. Coronavirus vaccines have been developed to prevent coronavirus infection and fight the pandemic. Since vaccines might be the only way to prevent and stop the spread of coronavirus. The World Health Organization (WHO) has already approved several vaccines, and many countries have started vaccinating people. Misperceptions about vaccines persist despite the evidence of vaccine safety and efficacy. Objectives: To explore the scientific literature and find the determinants for worldwide COVID-19 vaccine hesitancy as reported in the literature. Methods: PRISMA Extension for Scoping Reviews (PRISMA-ScR) guidelines were followed to conduct a scoping review of literature on COVID-19 vaccine hesitancy and willingness to vaccinate. Several databases (e.g., MEDLINE, EMBASE, and Google Scholar) were searched to find relevant articles. Intervention- (i.e., COVID-19 vaccine) and outcome- (i.e., hesitancy) related terms were used to search in these databases. The search was conducted on 22 February 2021. Both forward and backward reference lists were checked to find further studies. Three reviewers worked independently to select articles and extract data from selected literature. Studies that used a quantitative survey to measure COVID-19 vaccine hesitancy and acceptance were included in this review. The extracted data were synthesized following the narrative approach and results were represented graphically with appropriate figures and tables. Results: 82 studies were included in this scoping review of 882 identified from our search. Sometimes, several studies had been performed in the same country, and it was observed that vaccine hesitancy was high earlier and decreased over time with the hope of vaccine efficacy. People in different countries had varying percentages of vaccine uptake (28–86.1%), vaccine hesitancy (10–57.8%), vaccine refusal (0–24%). The most common determinants affecting vaccination intention include vaccine efficacy, vaccine side effects, mistrust in healthcare, religious beliefs, and trust in information sources. Additionally, vaccination intentions are influenced by demographic factors such as age, gender, education, and region. Conclusions: The underlying factors of vaccine hesitancy are complex and context-specific, varying across time and socio-demographic variables. Vaccine hesitancy can also be influenced by other factors such as health inequalities, socioeconomic disadvantages, systemic racism, and level of exposure to misinformation online, with some factors being more dominant in certain countries than others. Therefore, strategies tailored to cultures and socio-psychological factors need to be developed to reduce vaccine hesitancy and aid informed decision-making.

## 1. Introduction

Due to the coronavirus infection, the current pandemic is the topmost public health concern. With COVID-19 vaccines approved by the World Health Organization (WHO), the hope of overcoming the pandemic soon has increased. However, vaccines must be more widely accepted and used to end the pandemic [1]. The spread of the virus can also be mitigated by reaching herd immunity, but that takes more time [2]. Therefore, public awareness and well-designed campaigns promoting vaccination are essential to decrease the progression of COVID-19.

The WHO has already approved several vaccines and suggested getting the vaccine by majority of the population of a country as soon as possible to obtain herd immunity [2]. Despite evidence of the safety and effectiveness of vaccines, misperceptions about vaccines persist. Some people think that getting vaccinated can lead to temporary health impairments or long-term damage. *Vaccine hesitancy* is a complex phenomenon that affects people’s willingness to be vaccinated. Studies have shown that there is no single set of factors responsible for vaccine hesitancy. Instead, there is a wide range of contextual (i.e., communication and media, historical influence, religion, culture, gender, politics, geographic barriers), individual and group (i.e., personal, family experience with vaccination, beliefs, knowledge), and vaccine-specific factors (i.e., risk and benefit, costs) that can affect vaccine acceptance [1,3,4]. The cost of the vaccine may also affect willingness to be vaccinated because, in some countries, the cost is related to a person’s monthly income [5,6]. The factors affecting vaccination intention vary across countries, socioeconomic groups, demographic variables (i.e., ethnicity, gender), and types of infectious diseases [7].

Conspiracy theories and fake news propagating across social media have flourished during the COVID-19 pandemic [8]. In February 2020, when the pandemic rapidly grew worldwide, the WHO warned of an *infodemic*, a wave of fake news and misinformation on social media regarding COVID-19 [8]. After the approval of COVID-19 vaccines, misinformation about the vaccinations also started to disseminate quickly. The conspiracies frequently seen on social media include claims that COVID-19 vaccines change the human genome, that a microchip is implanted in the human body through the syringe, that the vaccination causes COVID-19 infections [9]. YouTube, Facebook, and Twitter announced working together to combat the problem. This misleading information may have affected the acceptance of COVID-19 vaccinations. Studies have also shown that rumors can have a negative effect on willingness to accept COVID-19 vaccines [5,10,11].

A *scoping review* is a relatively new literature review approach that follows evidence synthesis to provide an overview of the available research evidence. This overview answers broad questions rather than produces a summary answer [12,13,14]. This scoping review aimed to analyze published scientific literature that highlighted issues related to COVID-19 vaccine acceptance and hesitancy during the COVID-19 pandemic in various countries worldwide. After analysis, this scoping review summarized the factors that influence people’s willingness to be vaccinated. Exploring vaccination intention, along with underlying factors, helps us to understand public perception and attitudes towards vaccination. This study can be a starting point for further research about the factors contributing to regional and cultural variations in COVID-19 vaccine hesitancy.

## 2. Materials and Methods

PRISMA Extension for Scoping Reviews (PRISMA-ScR) guidelines were followed in conducting this review [15]. The detailed steps of the scoping review are listed in the following subsections.

### 2.1. Search Strategy

#### 2.1.1. Search Sources

Three bibliographic databases (Embase, PubMed, Google Scholar) were searched to retrieve studies related to COVID-19 vaccines hesitancy and acceptance. With regards to Google Scholar, the eligibility of 400 studies were checked by analyzing the first 20 pages. Because Google Scholar orders the retrieved studies by their relevance, retrieves a large number of studies, and does not have an advanced search tool (e.g., searching in titles and abstracts only) that can be used to run a precise and sensitive search [16,17]. In addition, both forward and backward reference list checking was performed to retrieve further studies. Our search was restricted to studies published between 19 February 2020 and 22 February 2021.

#### 2.1.2. Search Terms

Three distinct sets of search terms based on population, intervention, and outcome were selected. Population-related search terms contained coronavirus, intervention-related terms contained vaccine, and outcome-related terms contained hesitancy. To get more results, synonyms and similar terms were also added. In the end, the search strategy performed on the databases mentioned above formed these terms: (“novel coronavirus” OR “coronavirus 2019” OR “COVID 2019” OR “COVID19” OR “COVID-19” OR “SARS-CoV-2” OR “HCoV-19” OR “2019-nCoV” OR “severe acute respiratory syndrome coronavirus 2”) AND (vaccine * OR booster * OR inoculat * OR immune * OR immunization) AND (hesitan * OR reluctan * OR refus * OR accept * OR anti-vaccin * OR anti-vax).

### 2.2. Study Eligibility Criteria

Studies that used a quantitative survey to assess COVID-19 vaccine hesitancy and acceptance were considered for this study. The included studies were published in English between February 2020 and February 2021. Preprint articles, conference proceedings, and journal articles were included, and conference abstracts, proposals, literature review articles, overviews, dissertations, editorials, and commentaries were excluded. Restrictions on the modality of conducting the surveys, including sampling process, distribution, data collection, analysis, and the country where the data collection happened, were not imposed. Detailed information on inclusion and exclusion criteria is listed in Table 1.

### 2.3. Study Selection

The filtration process for selecting studies was conducted in three phases: identifying studies, screening the title and abstract, and full-text reading. Firstly, M.R.B. queried the databases and studies were retrieved in RIS (Research Information Systems) file format. All retrieved studies were uploaded to Rayyan software [18], a web-based systematic review tool that helps speed up the screening process. Duplicate studies were removed when using the software. In the second phase, the title and abstract of all the identified studies were separately reviewed by M.R.B. and M.S.A. Any disagreement between reviewers was resolved through consensus. Finally, the full text of the studies was screened by two reviewers, M.R.B. and M.S.A., and included in the review.

### 2.4. Data Extraction and Data Synthesis

For extracting data from the included studies, a data extraction form (Appendix A) was developed by authors M.R.B. and U.S. and was reviewed and approved by A.A.A.A. and Z.S. The data were extracted into an Excel sheet by two authors, M.R.B. and M.S.A., independently. Lastly, Z.S. reviewed the data extraction sheet.

The following data fields were extracted from the included studies: article type, the aim of the study, study design (e.g., quantitative, qualitative, and mixed), survey duration, data related to population (e.g., number of participants, gender, education, language, religion, location, and occupation), reasons for hesitancy and acceptance, and key findings of the study. A narrative approach was used to synthesize the extracted data.

## 3. Results

### 3.1. Search Results

The most popular three electronic databases (Embase, PubMed, and Google Scholar) were chosen to conduct the search process, and the search terms were applied to it. The search results retrieved 343 studies from Embase, 550 studies from PubMed, and 400 studies from Google Scholar, resulting in 1293 studies. The search results included books, journals, letters, commentary, editorial reviews, review papers, and conferences that needed to process with a further screening. The scoping review process is shown in Figure 1.

Firstly, 411 (31.786%) duplicate items were removed from the resulting studies, and 882 (68.21%) were kept for abstract and title screening. By applying inclusion and exclusion criteria through the abstract and title screening, 93 (10.54%) papers met the eligibility criteria, and the other 789 (89.46%) articles were excluded. Next, a full-text screening was conducted on the remaining 93 papers. While reading the complete text, nine papers were removed because the authors predicted COVID-19 vaccine hesitancy using vaccine hesitancy survey results of another disease (e.g., Influenza) and did not perform any survey for COVID-19 vaccine perception. In addition, two more papers were removed during the final screening of the included papers because the journal that published them withdrew them. This withdrawal happened because the authors duplicated one study and published it in two different journals. In the end, 82 papers were included in this scoping review. Appendix A contains the complete list of included articles.

### 3.2. Characteristics of Included Studies

#### 3.2.1. Summary of Included Studies

The included studies were conducted from February 2020 to February 2021. Table 2 shows the number of studies performed before and after vaccine approval for each country. It also classifies the publications based on the target population. Figure 2 shows the number of studies performed per month for various countries. The highest number of studies (*n* = 16) were performed in March 2020, where the maximum number (*n* = 4) was conducted in France. The second-largest number of studies (*n* = 13) were performed in May 2020, where six studies were conducted in the USA. Next, 11 studies were conducted in June 2020 and September 2020. Every other month, several studies were conducted in different regions of the world.

80 studies were conducted in 2020, and only two studies were performed in 2021. The highest number of studies were carried out in the USA (*n* = 23). In Italy, 10 studies were conducted; eight studies were conducted in France and China, and six studies were conducted in the UK. In both Australia and Hong Kong, four studies were performed, in both Canada and Turkey two studies were conducted, and in the remaining countries (Japan, Saudi Arabia, Ireland, Qatar, Poland, Kuwait, Congo, Jordan, Greece, and Poland) one study was conducted in each. One study included all 26 European countries in the survey. Another study included 19 countries from around the world. Among the 23 studies in the USA, 16 studies were conducted for the general population, two studies were conducted for students, three studies were conducted for healthcare workers (HCWs), and two studies were conducted for people suffering from serious health issues. Among the 10 studies in Italy, seven studies included the general population. Students, HCWs, and people at high risk were included in one study each. Among the eight studies in China, seven studies were based on the general population, and the remaining study included HCWs. Most of the studies were carried out in various countries and included the general population, highlighting the widespread perception of vaccination.

Of the studies included in this review, 77.77% (*n* = 63) were published in journals, 13.58% (*n* = 12) were preprint, and 8.64% (*n* = 7) were published in conference proceedings. Most of the surveys (*n* = 80) were performed before approval of the COVID-19 vaccination. Papers were classified into two categories: BV—before vaccine approval, from February 2020 to 15 November 2020, and AV—after vaccine approval, from 15 November 2020 to February 2021.

#### 3.2.2. Measurement Tools

COVID-19 vaccine hesitancy and vaccine acceptance were measured based on demographic variables and control variables. *Demographic variables* include age, gender, level of education, profession, ethnicity, population size, and monthly income [19]. *Control variables* are defined as those in a research study that scientists try to hold constant [20]. If a control variable changes during an experiment, the correlation between the dependent and independent variables may be invalidated. For example, in some studies, religiosity and political affiliation were selected as control variables but were not measured. This choice was made because, if political affiliation changes, other variables, such as ethnic minority, might be impacted.

The responses of participants were collected in various ways. Some participants used a 7-point Likert scale that ranged from *strongly disagree* to *strongly agree* [21,22,23]. Others used a 5-point Likert scale that ranged from *strongly agree* to *strongly disagree* but did not use *partially agree* and *partially disagree* [19,24].

The questionnaire sets were prepared by following the perceived health belief model. The health belief model is a conceptual framework designed by researchers to evaluate perceptions and attitudes towards vaccination [25]. The questionnaires were categorized into contextual influences, individual and group, and COVID-19 vaccine-specific effects. Most of the surveys were distributed through online platforms such as Google Forms, Microsoft Forms, SurveyMonkey, and Qualtrics.

Researchers applied different techniques to identify the determinants of COVID-19 vaccine acceptance or hesitancy. Some of them used univariate, bivariate, and multivariate associations with Pearson’s correlation [4]. Others used linear regression [26] to explore which variables predicted vaccination intention. Logistic regression was also used on the health belief model to assess the associations of demographic factors [27]. In addition, some researchers evaluated the model performance and computed area under the receiver operating curve (AUC). Lastly, they applied bootstrap resampling for internal validation and model optimization [24].

### 3.3. Worldwide COVID-19 Vaccine Acceptance and Hesitancy Rate

This section discusses the included studies and separates the results of COVID-19 vaccine data by country. In case of multiple studies for same type of population in any given country, the latest study results are shown in Table 3. For example, 16 studies were conducted to measure the general population’s willingness to vaccinate in the USA. Study conducted in [28] is the latest study to measure vaccine acceptance and hesitancy of the general population in the USA. Therefore, Table 3 shows the results of this study. The complete results documented on the data extraction sheet can be found in Appendix A.

Studies [5,11,19,24,26,27,28,29,30,31,32,33,34,35,36,37,38,39,40,41,42,43,44] measured COVID-19 vaccine uptake and hesitancy rates for the USA population. Early in May 2020, a study was conducted among the general population of the USA, where authors found that 31.13% of Americans did not intend to vaccinate against COVID-19 when a vaccine became available [5]. Later, in July 2020, another study [32] was conducted among the general population, in which it was observed that public attitudes towards vaccination improved, and there was a 50% to 70% association with a higher probability of choosing to vaccinate. Vaccination intention was measured to be 79% for the general population [28]. Another study measured that, for HCWs in the USA, vaccine uptake was 45% and vaccine hesitancy was 24% [11]. Among students in the USA, willingness to vaccinate was found to be 60.6% [26]. At the same time, people with a critical health condition, such as HIV, were found to be 34% hesitant to the COVID-19 vaccine [33].

Several studies [3,20,21,45,46,47,48,49] measured potential COVID-19 vaccine acceptance and hesitancy rates for people in Italy. The latest study [3] found that 15% of the general population in Italy would most likely refuse the vaccine, whereas 26% would be hesitant. Willingness to vaccinate was measured to be 83.2% among the HCWs in Italy [47], and students’ willingness to vaccinate was measured to be 86.1% [21]. Vaccine acceptance among people at-risk because of critical health conditions was measured at 86% [46].

Studies [6,22,50,51,52,53,54,55] measuring public attitudes towards COVID-19 vaccination in China showed that general population hesitancy was reported to be 25% in March 2020 [50]. Afterwards, vaccine hesitancy among the general population reduced to 10.9% [6], and, in September 2020 unwillingness to vaccinate was measured at 8.2% [22]. A total of 76.4% of the HCWs in China were willing to vaccinate [50].

Among the included studies [4,23,56,57,58,59,60] measuring the willingness to vaccinate in France, intention to refuse the COVID-19 vaccine increased over time. During the first coronavirus wave in May 2020, vaccine hesitancy was measured 28.5% among general population [56]. Over time, vaccine hesitancy increased to 32.8% in July 2020, 39.0% in August 2020, and 47.9% in September 2020 [56]. The willingness of HCWs in France to vaccinate was measured to be 48.6% [59] and for patients suffering from cancer diseases, willingness to vaccinate was measured to be 53.7% [23].

Six studies [61,62,63,64,65,66] conducted in the UK measured public perception towards COVID-19 vaccination. In April 2020, 18.9% of respondents stated that they were unsure, and 7.2% of respondents said they did not want to get vaccinated [66]. In September 2020, another survey was performed in which 71.7% of the general population were willing to vaccinate, 16.6% were unsure, and 11.7% were found to refuse the COVID-19 vaccine [65]. Finally, the willingness of high-risk people (i.e., people having serious health diseases like cancer, HIV, heart diseases) to vaccinate was measured at 86% in the UK [63].

Studies [67,68,69,70] measuring the willingness of Australians to vaccinate ranged from 76 to 86% [70]. In contrast, another study reported that if the COVID-19 vaccine was made available, 65% of respondents were willing to get vaccinated, and 27% were hesitant [67].

Three studies [25,71,72] measured vaccine hesitancy in Hong Kong. The vaccine intention for healthcare workers was 69% [71], while the general population was 57.8% hesitant [25]. In addition, students in Hong Kong were 40.6% hesitant to vaccination [72].

In Canada, 20% of the general population was unwilling to get the COVID-19 vaccine [73]. In comparison, 65% of caregivers intended to get vaccinated [74]. In Turkey, 31% of people were hesitant, and 54% were willing to vaccinate when the COVID-19 vaccine became available [62]. Another study reported that 43% of the healthcare personnel in Turkey were hesitant to get the COVID-19 vaccine [75]. On the other hand, a total of 65% of the general population in Ireland was willing to vaccinate [61]. Meanwhile, 73.9% of people in Denmark and Portugal were willing to accept the COVID-19 vaccine, 18.9% were unsure, and 7.2% refused to get vaccinated [76].

During the COVID-19 pandemic, general population willingness to vaccinate was reported as: 74% in Scotland [77], 37% in Poland [78], 78.3% in Indonesia [79,80], and 57.7% in Greece [81]. Several studies were conducted in the Gulf Cooperation Council (GCC) countries. Vaccine hesitancy among the general population in Qatar was measured at 19.8%, and vaccine refusal was measured at 20.2% [82]. A total of 35% of the general population in Saudi Arabia was hesitant about the COVID-19 vaccine [83]. Studies [84,85] reported that 53.1% of the general population in Kuwait was willing to vaccinate. Two studies were performed in Jordan [85,86], where the latest study [86] showed that 29.1% of the general population was willing to accept the vaccine when it became available. The survey in [87] included 19 countries worldwide, and the survey in [63] included 26 countries from Europe. In these studies, the authors measured public willingness to vaccinate in each country. Lower vaccine intention was reported in Congo, where 28% of HCWs were willing to vaccinate [10].

### 3.4. Determinants of COVID-19 Vaccine Hesitancy

There are three main types of factors that influence vaccine hesitancy or acceptance: (i) demographic factors (i.e., education, income, ethnicity), (ii) environmental factors (i.e., policies, media), and (iii) vaccine-specific factors (i.e., vaccine efficacy, safety) [74]. Determinants of vaccine hesitancy are specific to the context and are presented separately. Therefore, it is important to understand and acknowledge the interrelatedness of the factors [88]. We extracted *Reasons* for vaccine hesitancy for vaccine-specific and environmental factors, which can be found in Appendix A.

The most common factors that influence vaccination intention are described in Table 4. In Table 4, determinants affecting vaccine hesitancy, the papers the determinants were found in, the places the studies were conducted, the education levels of the population, and the occupations of the population are described. When Table 4 is analyzed, the relationship among determinants and other demographic factors like education, occupation, and the study’s place can be seen.

Vaccine safety and efficacy were found to be major concerns among people of all occupations. *Vaccine safety* determines the risk associated with vaccine benefit-risk profile changes and can anticipate coincidental events [89,90]. *Vaccine efficacy* determines the percentage of vaccine effectiveness against COVID-19 [41]. Public concerns regarding vaccine safety and efficacy were found in several studies [2,3,5,20,29,32,43,46,54,55,60,65,69,71,83].

Another finding is that some people were unwilling to vaccinate due to the side effects of vaccines, as reported in [11,19,25,29,40,42,53,54,68,77,79]. The COVID-19 vaccine is new, and it takes time to determine the vaccine’s short-term and long-term side effects [55].

Willingness to vaccinate was found to be associated with the perceived risk of being infected with the COVID-19. Younger people were more confident about being at less risk of being infected by the COVID-19 [20,27,40,43,53,54,59,64,91]. The perception of young people is also correlated with the concept of herd immunity of the body and knowledge about SARS-COV2 [22,46].

The conflict between vaccination science and religious people exists due to conspiracy theories about morality. Because of this, some religious people reject or delay vaccination [11,56,76,92]. This report predicted that religiosity strongly predicts anti-vaccine beliefs [56,76].

Due to a lack of health insurance or financial resources that might be necessary to have access to the vaccination, people were unwilling to vaccinate [5,6,39,55,80]. Moreover, the ability to spend money on health was often dependent on monthly income, creating health inequalities and increasing the vaccine hesitancy rate in certain countries [5,6].

Inconsistent messages from health organizations lead to hesitation in making decisions about vaccination. Throughout the coronavirus pandemic, mistrust in healthcare systems has increased [20,25,26,37,39,66,93]. Because of this, a vast number of people intend to postpone getting vaccinated.

Moreover, due to mistrust of the government, the general population was concerned about the vaccination information provided by government organizations, resulting in vaccine refusal [4,30,40,41,87].

Healthcare workers and the general population assumed that the vaccines were developed rapidly, might not pass a reasonable trial period, and were created without considering vaccine safety issues. Thus, they wanted to delay their vaccination to ensure effectiveness [35,40,69,79,84].

Many people use social media rather than traditional sources (newspaper, TV news) to find vaccine-related information. Anti-vaccine groups are active on social media and spread misinformation, which also influences willingness to vaccinate [32,61,63,69,81,85,93].

Past experiences with vaccine side effects were also associated with COVID-19 vaccine hesitancy [60,69,89]. People who have had other vaccines such as Influenza, Hepatitis-B, or Polio might have suffered from short-term side effects, making them unwilling to get vaccinated.

Demographic factors such as duration of the survey, population size, level of education, occupation, place of study, language, and religion are interrelated with each other and they influence the decision to vaccinate [34,42,45,62]. In addition, vaccine approval due to political pressure without proof of vaccine safety and efficacy was also reported as an issue that led to people refusing the COVID-19 vaccine [22,28,88].

Throughout history, ethnic groups (i.e., Black) have been the victims of systemic and institutional racism and discrimination. They have been used as a target population for vaccine trials [94]. This scenario undermines the trust level upon vaccination. From several studies [27,42,44], it has been observed that Blacks are unlikely to get the COVID-19 vaccine.

Trust in vaccination is also associated with the vaccine manufacturer [54,58,72,75,95]. However, a low percentage of people have confidence in the manufacturing companies developing safe and effective vaccines. The source of the COVID-19 vaccine might affect the perceived safety and effectiveness. It was assumed that vaccines manufactured in Europe or America were safer than those made in other countries [75].

Lockdowns and other precautionary measures, such as face masks, have reduced the number of infections [46]. Because of this, studies have found that some people became less interested in getting vaccinated [54]. It has also been found [27,36,38,43,91] that the inconsistent risk messages regarding COVID-19 reduced the intention of vaccine uptake. Moreover, the anti-vaccination movement on social media [31,39,64,79,91,92] has made a significant impact on people’s perception about vaccination.

Another reason for hesitancy is that some people prefer natural immunity [73]. This preference can stem from not trusting the scientific enterprise of medicine or not believing in it entirely [37]. In addition, some people want to know about the components and manufacturing process of the COVID-19 vaccine, which is not often revealed to the public. The lack of information provided by the manufacturers was also a reason for some people to be hesitant to get the vaccine [74,86,87,91].

### 3.5. Vaccine Hesitancy among Ethnic Minorities

Several studies measured the willingness of vaccine uptake among ethnic minorities. Research on previous vaccination programs reported a lower vaccination acceptance rate among ethnic minorities in the UK [21]. The survey performed in [21] showed that the vaccine acceptance rate among the White ethnic group was 84.5% and among the Black minority was 14.9%. Another study [62] reported that White study participants in the UK were twice more likely to accept vaccines than other ethnicities.

Studies [28,41,42,74] measured vaccination intention among American Indian, Asian, Black or African American, Hispanic or Latino, Native Hawaiian or Pacific Islander, White, and other minorities. In the USA, vaccine coverage among African Americans was lower in number compared to other race groups [74]. Another study [42] reported that Blacks were significantly less likely than non-Blacks to get vaccinated.

Limited research has been performed to identify the barriers to COVID-19 vaccine uptake among ethnic minority groups. The determinants found in the studies are stated below.

In general, ethnic minorities have less trust in government facilities due to institutional racism such as discrimination in education, employment, and housing and injustice in politics [96]. This phenomenon leads to Black and Hispanic minorities having a degraded willingness to be vaccinated [11].

Both Black and Hispanic minorities were also less willing to be vaccinated because they have less trust in government and healthcare systems [11]. In addition, due to the high rate of death among Black minorities occurred due to COVID-19, they were less willing to vaccinate [5,42].

Healthcare organizations have used ethnic minorities to perform experimental vaccine trials and other medical experiments [42]. Also, due to COVID-19, the highest burden of death and illness have been found among ethnic minorities [5]. Therefore, since these experiments have caused a high death rate and need for hospitalization among minorities, COVID-19 seems to be taking a more significant toll on minorities. The health inequalities and structural and institutional racism have led to minorities mistrusting in vaccination programs [44].

Anti-vax groups are active in delivering messages to Black Americans [96]. Moreover, conspiracy theories are relatively high among minorities. These factors negatively influence vaccine perceptions and lead to a lower acceptance of vaccines [21,62].

Ethnic minorities also believe that they are less likely to get infected by COVID-19 because their natural immune system is stronger than the virus. Since they have a lower perceived risk, they have a lower intention of getting vaccinated [54].

## 4. Discussion

### 4.1. Principal Findings

Vaccination, maintaining social distance, wearing face masks, and using sanitization, are critical measures to mitigating the spread of SARS-CoV2. Several pharmaceutical companies have developed COVID-19 vaccines, and many countries have already started vaccinating people. However, vaccination intention depends on demographic characteristics and partisanship, vaccine knowledge, perceived vulnerability to COVID-19, risk factors of COVID-19, and politics [27,42]. A study [87] reported that vaccination intention also varies from country to country.

From the findings, it has been observed that a higher vaccine acceptance rate was reported (70–80%) among the general populations of the USA, China, UK, Indonesia, Denmark, Scotland, and Poland. At the same time, the willingness to vaccinate the general populations in Italy, France, Australia, Turkey, Saudi Arabia, Greece, Kuwait, and Qatar was of a moderate range, 45–70%. On the other hand, lower vaccine uptake rates (30% to 45%) were recorded in Hong Kong, Poland, and Jordan.

A study [5] has reported that Americans are more likely to get vaccinated because of their higher probability of being infected by COVID-19. Similarly, it was found that the behavior of people in Italy towards vaccination has positively changed due to a higher infection rate in the country [3].

Vaccine hesitancy was different among different ethnic groups (e.g., Black) due to their racialized and minoritized communities with distinct cultures and social norms [77]. In addition, lower trust in government authorities and healthcare systems led minorities to have a lower intention of getting vaccines.

High acceptability of vaccination was found among some high-income people because they had a higher perception of the risk of catching the virus [34]. In addition, these people were aware of the vaccine’s effectiveness, which helped them accept the vaccine more easily.

Widespread misinformation communicated through social media was responsible for degrading vaccine acceptance rates [21]. Unfortunately, some people strongly believe whatever they see on social media and share it among their friends and colleagues without knowing the accuracy of the information. This sharing has led to the spread of misinformation and reduced faith in the vaccine.

Other reasons for a low willingness to vaccinate were that the vaccines were very new; there was no proof of efficacy through rigorous testing, and the vaccine’s safety was questionable [41]. Many people talked about these reasons and how they were less willing to vaccinate because of them.

The HCWs are the front-line workers combating COVID-19. The willingness to vaccine uptake depends on the general trust of the public health system managed by medical providers and government agencies [3]. Therefore, vaccination among HCWs is considered a high priority issue. The average percentage of HCWs vaccine uptake for China, the UK, Canada, and Italy was more than 60%. However, for the USA, France, and Congo, the HCWs vaccine uptake rate was less than 48%. Reasons for low vaccine uptake among HCWs in the USA were the belief in potential side effects and insufficient trust in regulatory authorities and the government regarding vaccine development and distribution.

Vaccine efficacy has a significant impact on willingness to uptake vaccines. Studies have observed that the rate of COVID-19 vaccine acceptance changed with reported vaccine effectiveness changes. A survey conducted by [41] showed that people’s vaccine acceptance rate changed to 40.58%, 47.35%, and 56.70%, while the vaccine efficacy rate was 50%, 75%, and 99%, respectively. Another study conducted by [6] showed that the vaccine acceptance coefficient varied with it being 3.138 (*p* < 0.001) in the case of 90% vaccine effectiveness and reducing to 1.416 (*p* < 0.001) in the case of 70% vaccine effectiveness. Since general people believe that vaccine efficacy depends on prolonged testing [71], they were hesitant to take the COVID-19 vaccine. This belief was especially seen in older people (more than 65 years old) [46]. People’s attitudes towards the COVID-19 vaccine tend to change with time intervals because they want to observe more data that has been collected after people have been vaccinated [54,70]. According to the research, if the vaccine efficacy increases and protects people for a more extended amount of time, the vaccine acceptance rate will also increase [28].

Vaccine acceptance rates were found inversely proportional to vaccine side effects. The perceived risk of getting infected by COVID-19 during vaccination appeared to be a significant predictor of vaccine refusal [83]. Older people and those who are more susceptible to clinical complications were found less interested in getting vaccinated because they were more concerned about the side effects of the COVID-19 vaccine [47,85]. Other people believe that, after COVID-19 vaccination, the probability of side effects is high, so they are less willing to be vaccinated [97]. Thus, it is an unprecedented challenge for public health authorities to build trust among the general population and those who have previously had bad experiences with medical providers and government agencies [27].

Both scoping and systematic reviews were conducted to explore the COVID-19 vaccine hesitancy throughout the world [93,98,99,100,101]. Study [93] performed a systematic review and reported vaccine acceptance and hesitancy from 30 studies. Study [98] included total 22 papers in their scoping review and reported the findings of vaccine hesitancy. Whereas study [99] performed a systematic review and reported vaccine hesitancy and acceptance from low-income and high-income countries. Study [100] included 97 papers in their scoping review after filtering process and discussed about the determinants of vaccine hesitancy in the high-income countries. Another scoping review was performed in [101] and included 66 studies to analyze vaccine hesitancy among nurses and pharmacists. The search results and reported vaccine hesitancy published in other scoping review papers were relevant to our scoping review. The common characteristics found all other scoping review was that vaccine hesitancy decreased over time and vaccination intention increased. Our scoping review differs with other in finding intervention and reporting the result. While comparing with other scoping review papers, we identified the included studies that surveyed population to report results. We have figured out the change of COVID-19 vaccine hesitancy and acceptance over time. Moreover, we also analyzed the underlying determinants of vaccine hesitancy during the COVID-19 pandemic. In our study, we categorized the included study results into four categories: general population, healthcare workers, students, and critical health conditions. Each category represents a portion of the population, acceptance rate, rate of hesitancy, place where the study was conducted, occupation, and education level. Since this study summarizes a wide range of factors that are reported in literature, these results are more useful for future studies.

The included studies were published before observing the efficacy level and potential short-term and long-term side effects of the COVID-19 vaccine. People were most concerned about the after-effects of the vaccine, and their vaccination intention was dependent on these concerns. Later, while the paper was in revision, several studies [102,103,104,105] were performed to analyze the after-effects of the vaccine. A report published in [102] stated that the COVID-19 vaccination strategy significantly reduced the hospitalization rate of people of all ages and, more specifically, the hospitalization rate of people 80 years or older reduced by 80%. In addition, study [104] demonstrated that prompt vaccine distribution significantly reduced coronavirus transmission. However, common short-term side effects after vaccination that were reported in [105] were injection site pain (89.8%), fatigue (62.2%), headache (45.6%), muscle pain (37.1%), and chills (33.9%). There is ongoing, continuous research to determine the long-term side effects of the COVID-19 vaccine. In the future, gathering evidence on vaccine safety might increase the vaccine confidence level.

### 4.2. Strengths and Limitations

#### 4.2.1. Strengths

In this scoping review, we considered those studies where the authors performed a quantitative survey to measure hesitancy or willingness to accept the COVID-19 vaccine. Studies with only theoretical data analysis and no survey results were excluded. The included studies estimated vaccine hesitancy rates from different countries throughout the world, where the target population was those 18 years or older with any profession. While, at times, there were multiple studies conducted in the same nation, we considered the latest study result to be the final vaccine acceptance rate. To determine reasons for hesitancy, we discussed the significant determinants described in the studies. This study provides the percentage of COVID-19 vaccine uptake and vaccine hesitancy before and after vaccine approval and identified factors that influence vaccination intention.

#### 4.2.2. Limitations

Firstly, we considered only three databases (Google Scholar, PubMed, and Embase) for our search. There are other databases such as Scopus, PsycINFO, PMC, and NCBI, which we did not explore. Secondly, the included studies may be biased with the publications since some of the papers were preprint. Preprint paper are not peer reviewed and so methodology and results might change for final version of accepted manuscript. While finding the determinants of vaccine hesitancy, we could have categorized each determinant by country, making it easier to understand the underlying factors affecting COVID-19 vaccine hesitancy for each country. Vaccine hesitancy and vaccine acceptance repeatedly change over time as vaccine efficacy and side effects are observed, so the findings of this study may not be valid after a certain period. In this study, our literature review contained studies between February 2020 to February 2021, which provides a comprehensive overview of vaccine acceptance and hesitancy before the mass vaccination of people was started. In several months, public attitudes towards vaccination may have changed because more worldwide vaccination data has been observed. Therefore, these results might not be valid for measuring vaccination intention after a certain period, and further analysis may be needed to identify changes in preference.

### 4.3. Practical and Research Implications

#### 4.3.1. Practical Implications

Some people are highly influenced by social media and contacts (friends, family, and colleagues) [44]. Conspiracy theories (e.g., stating that coronavirus was manufactured in a laboratory) derived from these sources negatively affect vaccination intentions [3,56]. To make people aware of the COVID-19 vaccine, social and educational campaigns are needed. People should be motivated to understand how dangerous the disease is for them and those living around them [73]. Studies have reported that older people are hesitant to get the vaccine because they do not feel safe about what may happen after vaccination, and younger people are hesitant because they do not think they will be affected by COVID-19 [81]. Both healthcare professionals and religious leaders can play an essential role in positively shaping people’s beliefs about vaccination because they influence society [78,92]. To increase people’s awareness about the COVID-19 vaccination, there need to be frequent social campaigns that highlight the vaccine’s usefulness.

People who have a higher level of education are more concerned about the risk of catching COVID-19 and have a greater willingness to get vaccinated [34]. They also have access to multiple information sources (e.g., health agencies to personal networks to social media), which is vital in predicting vaccine acceptance [72]. Educationalists tend to engage themselves in safeguarding and raising positive awareness regarding vaccination [61]. Since many students are part of communities, public health organizations should also increasingly focus on this population to increase COVID-19 vaccine coverage [43].

It has been observed that people whose monthly incomes are high are more willing to get vaccinated [54]. In contrast, those with low monthly incomes are less willing to get vaccinated due to a lack of health insurance and vaccine prices [5]. The findings show that high-income people are more aware of the negative consequences of COVID-19 and want to get the vaccination to maintain their wellbeing [77]. So, affordable vaccine price and exemption of health insurance might increase the vaccine acceptance rates.

In addition, the public is aware of the vaccine’s effectiveness. Therefore, increasing vaccine coverage requires convincing evidence and clear communication regarding the safety and effectiveness of the COVID-19 vaccine [76].

#### 4.3.2. Research Implications

Vaccination during the COVID-19 pandemic is a high-priority task because vaccination mitigates the spread of the coronavirus. Therefore, it is essential to understand public attitudes and perceptions towards vaccination to fulfill immunization targets. This scoping review might be the initial step to knowing how to improve vaccine coverage throughout the world. This paper concludes with the statistical results of vaccine uptake and vaccine hesitancy of countries according to population type. These results could be helpful for vaccine campaign programs. Additionally, this paper highlights the primary reasons that affect vaccination intention. Identifying the reasons that affect vaccination intention for each specific country might allow programs to help reduce vaccine hesitancy and increase vaccination rates during the pandemic.

This scoping review is limited to a specific period, and public perception was measured worldwide. However, public attitudes might change as the time to observe vaccine efficacy and side effects increases. Therefore, continuous research of public attitude variations and perceptions needs to be conducted. This research will be an initial guideline to understanding the primary concept of public attitude towards vaccination and change over time.

In the included studies, it was observed that most of the studies surveyed the general population. However, public attitudes vary by region, gender, age, occupation, and level of education. Therefore, a vast amount of research using population-specific survey questions should be conducted. This research will help get more precise results about vaccination intention. Furthermore, much research needs to be conducted to identify the interrelation between determinants of vaccine hesitancy and demographic variables (e.g., gender, age, education, occupation).

## 5. Conclusions

This scoping review focused on the determinants of vaccine hesitancy and reported the rate of vaccine hesitancy and willingness to vaccinate in different countries and different type of population. It has been observed that vaccine hesitancy and willingness to vaccinate vary from country to country and change with the passage of time. The hesitancy rate of some countries was relatively high, whereas, in other countries, vaccine uptake was high, and hesitancy was low. Many factors affect the vaccine hesitancy of a nation. The most common characteristic of willingness to vaccinate was vaccine efficacy. However, conspiracy theories and using online platforms to spread misinformation worldwide have also hindered the vaccine’s acceptance. People share their own belief through social media. So, monitoring the trends in social media might be useful to understand the vaccination intention of general people. Healthcare authorities are the most trusted source of information regarding vaccination. Therefore, they should take steps to make people aware of the vaccine’s effectiveness. These steps can be taken by implementing online social campaigns that deliver trusted news, providing free consultations related to vaccine concerns, increasing transparency of the vaccine manufacturing process, and ensuring vaccine safety and efficacy. Along with the delivery of trusted news, the ease of vaccine access needs to be increased to motivate more people to get vaccinated to acquire herd immunity.

## Figures and Tables

**Figure 1 vaccines-09-01243-f001:**
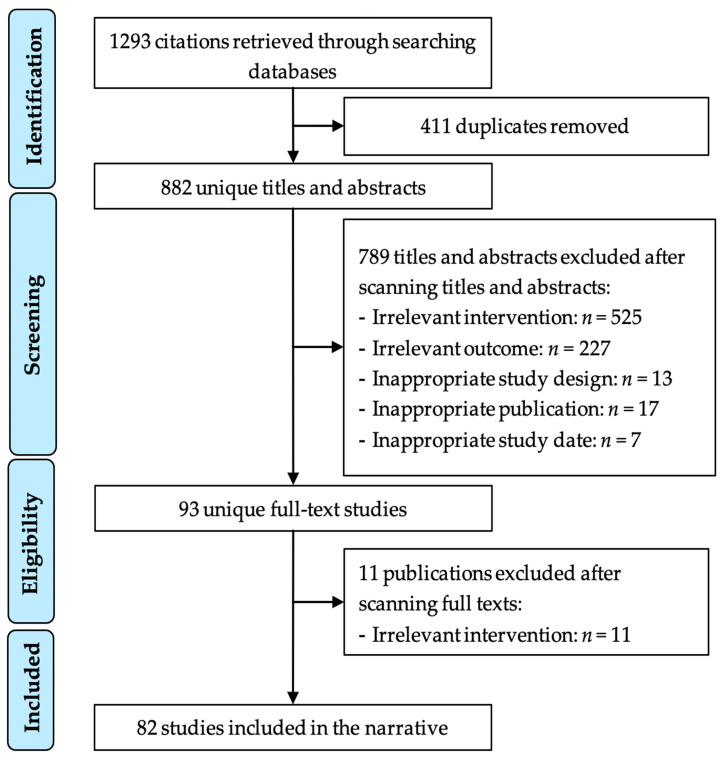
Study selection process.

**Figure 2 vaccines-09-01243-f002:**
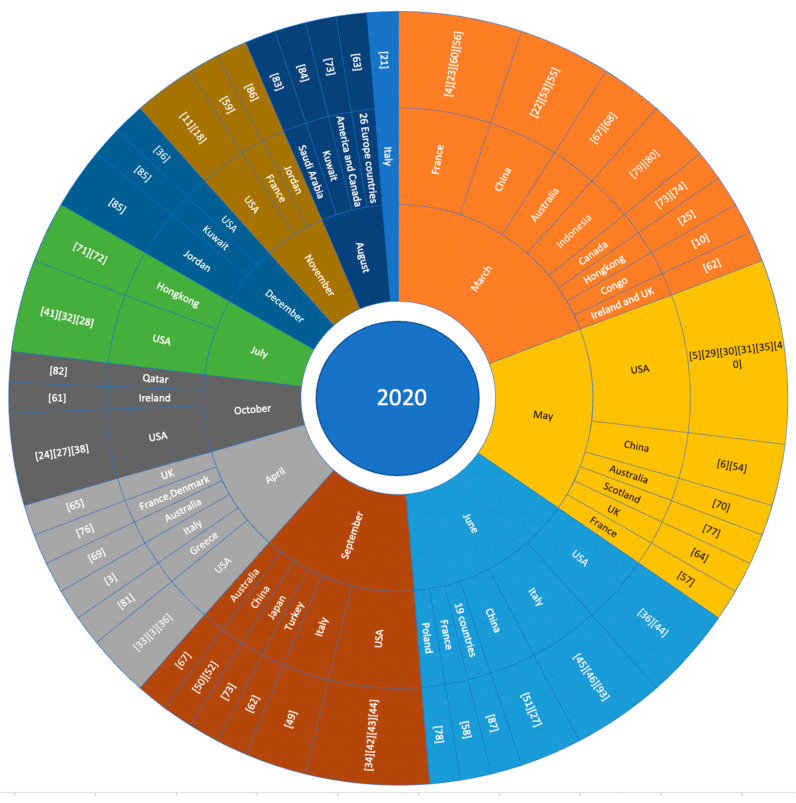
Studies conducted in 2020.

**Table 1 vaccines-09-01243-t001:** Inclusion and exclusion criteria.

Criteria	Specified Criteria
Inclusion	Intervention: quantitative approaches are used to identify COVID-19 vaccine hesitancy or vaccine acceptance in a specific way, such as questionnaires on the COVID-19 vaccine through online and offline surveys and statistics. Population: no restriction applied. Outcome: must include COVID-19 vaccine hesitancy explicitly or implicitly Type of publications: conference papers, journal papers, theses, reports, preprint Language: English
Exclusion	Study-related to other than COVID-19 vaccine hesitancy Studies having only deliverable or narrative results Type of publications: commentary, editorial review, newspaper article, overview, abstract, review paper Language: other languages

**Table 2 vaccines-09-01243-t002:** Characteristics of the included studies.

Country	No. of Studies		Target Population			
	BV (79)	AV (2)	General Population	Students	Health Workers	People at High Risk
United States	23	0	16	2	3	2
Italy	10	0	7	1	1	1
China	8	0	7	0	1	0
France	6	1	5	0	2	1
United Kingdom	6	0	5	0	0	1
Australia	4	0	4	0	0	0
Hong Kong	3	0	1	1	1	0
Canada	2	0	1	0	1	0
Turkey	2	0	0	1	1	0
Ireland	1	0	0	0	0	1
Poland	1	0	1	0	0	0
Japan	1	0	1	0	0	0
Saudi Arabia	1	0	1	0	0	0
Qatar	1	0	1	0	0	0
Oman	1	0	1	0	0	0
Jordan	1	1	1	0	0	0
Kuwait	1	0	1	0	0	0
Congo	1	0	1	0	0	0
Greece	1	0	1	0	0	0
Europe (26 Countries)	1	0	1	0	0	0
World (19 Countries)	1	0	1	0	0	0

BV—Before Vaccine approval, AV—After Vaccine approval.

**Table 3 vaccines-09-01243-t003:** COVID-19 vaccine coverage rate.

Country	Type of Population	Population Size	Vaccine Uptake (%)	Vaccine Hesitancy (%)	Vaccine Refusal (%)
USA	GP [28]	N = 1971, F = 51%	79	21	N/S
	HCW [11]	N = 8243, F = 87%	45	24	N/S
	Student [26]	N = 1062, F = 79.8%	60.6	15.1	24.3
	CRC [33]	N = 101, M = 77%	N/S	34	N/S
Italy	GP [3]	N = 1004, M = 49.1%	58.6	26	15
	HCW [47]	N = 968	83.2	16.3	N/S
	Student [21]	N = 735, F = 79.6%	86.1	13.9	N/S
	CRC [46]	N = 2267, F = 69%	86	13	N/S
China	GP [6]	N = 1236, F = 51.1%	80	10.9	8.1
	HCW [50]	N = 541, F = 60%	76.4	20	3.6
France	GP [56]	N = 4027	48.8	47.9	N/S
	HCW [59]	N = 2047, F = 75%	48.6	23	N/S
	CRC [23]	N = 999, F = 56.1%	53.7	N/S	N/S
UK	GP [65]	N = 3667, M = 50.1	71.7	16.6	11.7
	CRC [63]	N = 527, F = 57%	86	N/S	N/S
Australia	GP [69]	N = 1420	69	10	N/S
Hong Kong	Student [72]	N = 1200, F = 71.4	N/S	40.4	17.4
	GP [25]	N = 1200, M = 28.7%	42.2	57.8	N/S
	HCW [71]	N = 1205, F = 90%	63	N/S	N/S
Canada	GP [73]	N = 3674, F = 43%	N/S	N/S	20
	HCW [74]	N = 1541	65	N/S	N/S
Turkey	GP [62]	N = 3936	54	31	N/S
	HCW [75]	N = 1138, F = 72.5	N/S	43	N/S
Ireland	GP [61]	N = 1041	65	N/S	N/S
Denmark and Portugal	GP [76]	N = 7664	73.9	N/S	N/S
Scotland	GP [77]	N = 3436	74	N/S	N/S
Poland	GP [78]	N = 1066	37	N/S	N/S
Congo	HCW [10]	N = 613, F = 49.1	28	N/S	N/S
Greece	GP [81]	N = 1004, F = 49%	57.7	16.3	26
Indonesia	GP [80]	N = 1359, F = 65.7	78.3	N/S	N/S
Qatar	GP [82]	N = 7821, M = 59.4%	N/S	19.8	20.2
Saudi Arabia	GP [83]	N = 992, M = 34%	N/S	35	N/S
Jordan	GP [87]	N = 2173, M = 30.6%	29.1	N/S	N/S
Kuwait	GP [85]	N = 2368, F = 67.4%	53.1	N/S	N/S

GP—General Population, HCW—Health Worker, CRC—Critical Health Condition F—Female, M—Male, N/S—Not Specified.

**Table 4 vaccines-09-01243-t004:** Determinants of COVID-19 vaccine hesitancy.

Determinants	No. of Paper	Place of Study	Education	Occupation
Vaccine safety and efficacy	15	USA, China, Hong Kong, Australia, England, France, Qatar	Undergrad	HCW, full-time employee
Vaccine side effects	12	USA, China, Canada, Turkey, Kuwait	High school, secondary	Workers, employee, nurse
Individuals believe that they are at less risk to get infected by COVID-19	9	USA, Saudi Arabia, UK, Italy	High school to university	Employee
Religious beliefs	5	France, Denmark, Portugal, Germany	High school	Not specified
Price of vaccine and lack of insurance	5	China, Indonesia, USA	Primary school and high school	Private sector employee
Mistrust in healthcare	7	USA	College education	Student, employed
Mistrust in government	6	France, Ireland, Italy, USA	All level	All profession
The rapid development of a vaccine	5	Jordan, USA, UK	University level	Doctors, nurse, employed
Widespread misinformation in the social media	7	Greece, European countries, Jordan, Kuwait	High school	Student, employed, unemployed, retired
Past vaccine experience	3	Australia, France	Diploma	Health workers
Demographic influence	4	Turkey, USA, Italy	High school, bachelor	All profession
Political instability	3	USA	All level	HCWs, all profession
Racist and ethnic minority	3	USA	High school, bachelor	All profession
Trust in the vaccine manufacturer	5	China, Hong Kong	Primary to bachelor’s degree	HCWs, employee, student
Lockdown periods decrease the number of cases	1	Italy	High school	All level
Trust in natural remedies	1	America and Canada	All level	Full-time and part-time employee
Lack of information about vaccine	4	Saudi Arabia, Qatar, Kuwait, Jordan	High school to graduate	Employed
Inconsistent risk message from public health organization	4	USA, Canada, UK	All level	All level
Anti-vaccination movement	4	USA, Jordan, Europe	High school to undergrad	Employed

## Data Availability

Not applicable.

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
