# Peer review of "A Scoping Review to Find Out Worldwide COVID-19 Vaccine Hesitancy and Its Underlying Determinants"

_vaccines, 2021, doi:10.3390/vaccines9111243_

Round 1
Reviewer 1 Report
Estimated Authors of the paper "A scoping review to find out worldwide COVID-19 vaccine hesitancy and its underlying determinangs",
I've read with great interest your paper, with a substantially positive feedback.
In other words, I think that your paper may be only some step away from the acceptance by Vaccines. More precisely, I have the following requirements/suggestions:
- Search strategy: the choice to include only 389 studies from Google scholar is reasonable, but quite unclear. Why 389 ? The rationale behind such choice must be explained. Similarly, Authors should explain why they did scan only the first 20 pages of google results, possibly providing some references for such approach.
- Study eligibility criteria: the present paper was designed and the study performed several months ago. In the ever changing face of the COVID-19 pandemic this may be a significant limit, and must be somewhat addressed in the discussion section: the status of the vaccine hesitancy in the summer 2021 had radically changed from the winter months of 2020.
- Sections 3.2.1-3.2.2 etc. Authors have provided clearly designed tables, but some of the data (e.g. overall sampled population, representation of the reported studies on the overall collected data etc) should be included across the text.
- Page 8 and following: even though the summarized results are provided in great detail, the reported information sometimes lacks details on the percentages and crude figures; please make consistent (e.g. the studies conducted in [50-56] measured ... strongly hesitant" represent a valuale blueprint for this section.
- The topic of the minorities (e.g. the ethnic group (i.e. black...)) is of great interest, but it deserves larger details and space. Please expand it. On this regard...
- ... I think that the paper must benefit from an extensive revision of the English Text. Even though no significant mistakes may be found across the text, the phrasing is often awkward or affected by a very rigid lexical construction.
Reviewer 2 Report
Authors reviewed the possible link to covid-19 vaccine hesitancy. Authors need to intensively review the severe cases including deaths in covid-19 vaccinated humans. Even though it is not officially confirmed, thousands of death cases in the vaccinated humans were reported in many countries, for example, in USA, over 12,000 human deaths. Surely, these outcomes severely affect covid-19 vaccine hesitancy.
Round 2
Reviewer 2 Report
Authors responded to my concern.